# Intention to Inform Relatives, Rates of Cascade Testing, and Preference for Patient-Mediated Communication in Families Concerned with Hereditary Breast and Ovarian Cancer and Lynch Syndrome: The Swiss CASCADE Cohort

**DOI:** 10.3390/cancers14071636

**Published:** 2022-03-23

**Authors:** Mahesh Sarki, Chang Ming, Souria Aissaoui, Nicole Bürki, Maria Caiata-Zufferey, Tobias Ephraim Erlanger, Rossella Graffeo-Galbiati, Karl Heinimann, Viola Heinzelmann-Schwarz, Christian Monnerat, Nicole Probst-Hensch, Manuela Rabaglio, Ursina Zürrer-Härdi, Pierre Olivier Chappuis, Maria C. Katapodi

**Affiliations:** 1Department of Clinical Research, University of Basel, 4055 Basel, Switzerland; mahesh.sarki@unibas.ch (M.S.); chang.ming@unibas.ch (C.M.); 2Breast Center, Cantonal Hospital Fribourg, 1752 Fribourg, Switzerland; souria.aissaoui@genesupport.ch; 3GENESUPPORT, The Breast Centre, Hirslanden Clinique de Grangettes, 1224 Geneva, Switzerland; 4Women’s Clinic, University Hospital Basel, 4031 Basel, Switzerland; nicole.buerki@usb.ch (N.B.); viola.heinzelmann@usb.ch (V.H.-S.); 5Department of Business Economics, Health and Social Care, University of Applied Sciences and Arts of Southern Switzerland, 6928 Manno, Switzerland; maria.caiata@supsi.ch; 6Clinical Trials Unit, University Hospital Basel, 4031 Basel, Switzerland; mail@tobiaserlanger.ch; 7Oncology Institute of Southern Switzerland, EOC, 6500 Bellinzona, Switzerland; rossella.graffeogalbiati@eoc.ch; 8Institute for Medical Genetics and Pathology, University Hospital Basel, 4031 Basel, Switzerland; karl.heinimann@usb.ch; 9Research Group Human Genomics, Department of Biomedicine, University of Basel, 4031 Basel, Switzerland; 10Department of Medical Oncology, Hospital of Jura, 2800 Delemont, Switzerland; christian.monnerat@h-ju.ch; 11Swiss Tropical and Public Health Institute, University of Basel, 4123 Allschwil, Switzerland; nicole.probst@swisstph.ch; 12Department of Medical Oncology, Inselspital, Bern University Hospital, 3010 Bern, Switzerland; manuela.rabaglio@insel.ch; 13Department of Medical Oncology, Cantonal Hospital Winterthur, 8400 Winterthur, Switzerland; ursina.zuerrer@ksw.ch; 14Unit of Oncogenetics, Division of Oncology, University Hospitals of Geneva, 1205 Geneva, Switzerland; pierre.chappuis@hcuge.ch; 15Division of Genetic Medicine, University Hospitals of Geneva, 1205 Geneva, Switzerland

**Keywords:** family-based cohort, family invitation, LASSO, public health genetics, Tier 1 genetic syndromes, untested relatives

## Abstract

**Simple Summary:**

This paper presents important information for the implementation of cascade screening programs for hereditary breast and ovarian cancer (HBOC) and Lynch syndrome (LS). The study examined theory-based predictors of index cases’ intention to invite relatives to a family-based cohort, rates of cascade testing among relatives, and preferences of those who carry a pathogenic variant for patient- or provider-mediated communication of testing results to untested relatives. While index cases are equally likely to invite relatives of both genders, males are more likely to not respond to the invitation, especially for HBOC. Findings have implications for tailoring cascade screening programs.

**Abstract:**

Cascade screening for Tier 1 cancer genetic conditions is a significant public health intervention because it identifies untested relatives of individuals known to carry pathogenic variants associated with hereditary breast and ovarian cancer (HBOC) and Lynch syndrome (LS). The Swiss CASCADE is a family-based, open-ended cohort, including carriers of HBOC- and LS-associated pathogenic variants and their relatives. This paper describes rates of cascade screening in relatives from HBOC- and LS- harboring families, examines carriers’ preferences for communication of testing results, and describes theory-based predictors of intention to invite relatives to a cascade screening program. Information has been provided by 304 index cases and 115 relatives recruited from September 2017 to December 2021. On average, 10 relatives per index case were potentially eligible for cascade screening. Approximately 65% of respondents wanted to invite relatives to the cohort, and approximately 50% indicated a preference for patient-mediated communication of testing results, possibly with the assistance of digital technology. Intention to invite relatives was higher for first- compared to second- and third-degree relatives, but was not different between syndromes or based on relatives’ gender. The family environment and carrying pathogenic variants predicts intention to invite relatives. Information helps optimize delivery of tailored genetic services.

## 1. Introduction

Every year, more than 12,000 individuals are diagnosed with breast, colorectal, endometrial, and ovarian cancer in Switzerland [1]. Approximately 10% of breast cancer, 15–20% of ovarian cancer, and 5% of prostate cancer cases develop due to inherited germline pathogenic variants associated with hereditary breast and ovarian cancer (HBOC) syndrome [2]. Lynch syndrome (LS) accounts for 2–6% of colorectal and endometrial cancers, and other malignancies including ovarian, pancreatic, and gastric cancers [3]. Individuals with pathogenic variants associated with HBOC and LS often develop cancer younger than 50 years old and before screening recommendations apply [4,5]. The availability of counseling and testing for these two syndromes have made a significant contribution to cancer prevention and control. HBOC patients can benefit from personalized treatments, increased surveillance, chemo-prevention, and/or risk-reducing surgery, while LS carriers with colorectal cancer could consider subtotal colectomy or immune checkpoint therapy in advanced cases [5,6]. Carriers at reproductive age may also seek advice for prenatal diagnosis and assisted reproduction, including pre-implantation genetic testing.

Cascade genetic screening is the practice of identifying and offering testing to relatives of individuals known to carry pathogenic variants associated with autosomal dominant conditions, such as HBOC and LS. Cascade testing starts with first-degree relatives of index cases (i.e., first person in a family to be identified with a pathogenic variant) and then proceeds to second- and third-degree relatives, who have an a priori 50%, 25%, and 12.5% probability, respectively, for inheriting the respective cancer predisposition. Cascade testing identifies relatives with the familial pathogenic variant that require personalized cancer risk management, and relieves non-carriers from intensive cancer surveillance and prevention interventions [7]. Cascade screening for monogenic actionable (Tier 1) conditions can reduce the risk of adverse health outcomes in entire cohorts of relatives [8]. It is a cost-effective approach for identifying at-risk individuals, especially in young, unaffected relatives [9,10,11]. Monitoring variant-harboring families provides important information for planning adequate quantity and quality of services and long-term coordination of cancer care [12]. However, despite the benefits of cascade testing, studies report that a significant number of relatives remain unaware of the availability and benefits of genetic testing. Reviews report that uptake of HBOC- and LS-associated cascade testing among at-risk relatives is less than 50%, depending on the syndrome and the clinical settings [13,14]. Individual studies further report significant underutilization of cascade testing among at risk relatives, leading to significant missed opportunities for cancer prevention [15,16,17]. Reasons for these missed opportunities are important to understand in order to tailor interventions according to each syndrome and to important characteristics of the target population.

The Swiss Cancer Genetic Predisposition Cascade Screening Consortium was assembled in 2016 to promote research related to hereditary cancer predisposition and the CASCADE cohort, a family-based cohort targeting HBOC and LS variant-harboring families [18]. The specific aims of this paper are to describe HBOC and LS index cases in the cohort, characteristics of relatives potentially eligible for cascade testing, rates of testing, preferences for provider involvement in the communication process, and to explore predictors of intention to invite relatives to the cohort.

## 2. Material and Methods

CASCADE is an open-ended cohort designed to elicit factors that enhance carrier testing for HBOC and LS (NCT03124212) [18]. CASCADE targets index cases and extends the invitation to their relatives, to examine if they had carrier testing, if they also carry the familial pathogenic variant, and how they manage their risk for hereditary cancer. Data collection for the cohort occurs about 18–24 months apart. A follow-up survey was launched in April 2020 to 94% of respondents willing to participate in the cohort. This paper presents data collected from respondents enrolled between September 2017 and December 2021. The Ethics Committee of Northwestern and Central Switzerland and the ethics committees of clinical sites approved the study protocol (BASEC 2016-02052).

Index cases are recruited from eight oncology and genetics centers (university and cantonal hospitals and private clinics) in German-, French-, and Italian-speaking regions of Switzerland. Index cases are asked to recruit first-, second-, and third-degree relatives. Eligible are male and female adults (≥18 years old) who have been identified as carriers of HBOC- or LS-associated pathogenic variants, their relatives who did not have genetic testing (unknown status), those who were identified as carriers of the familial pathogenic variant, and true negative relatives (i.e., carrier testing excluded the familial pathogenic variant). Eligible are cancer patients and cancer-free individuals, residents of Switzerland at the time they enter the cohort, able to provide written consent, and can complete a survey either in German, French, Italian, or English. Excluded are children, carriers of variants of unknown significance, and non-biological relatives, because carrier testing does not apply. Excluded also are vulnerable individuals (i.e., critically ill) because they may not be able to follow recommendations for cancer surveillance.

Clinical sites identify index cases, i.e., older cases from clinic records and individuals newly diagnosed with HBOC- or LS-associated variants, determine whether they are eligible for the study, and initiate recruitment by mailing an invitation letter signed by the medical director. The recruitment process includes three mailing attempts. Index cases who are interested in the study return a signed consent and complete the baseline survey, which asks them to indicate all living first-, second-, and third-degree relatives older than 18, living in Switzerland, and who they are willing to invite to the cohort.

Research staff reviews the baseline survey of index cases and identifies all relatives that the index case is willing to contact and all relatives potentially eligible for cascade testing. In order to alleviate ethical concerns associated with contacting relatives without their explicit consent, index cases are asked to pass on an invitation letter signed by the study Principal Investigator. By signing and returning a written consent, relatives provide their contact information to the research team, demonstrating their willingness to participate to the study. Relatives receive a baseline survey and indicate if they are willing to invite further relatives. This additional recruitment step (relatives further inviting relatives) uses multiple contact pathways within the family network to capture as many individuals from these variant-harboring families as possible. True negative relatives receive special instructions to not invite their offspring. If necessary, index cases and relatives receive a six-week reminder to complete and return their baseline survey. Respondents receive an annual newsletter, and once a year one participant is randomly chosen to receive a 300 Swiss Francs gift card (approximately 325 U.S. dollars) as incentives to participate and remain in the cohort.

Surveys for index cases and relatives include almost identical scales, except when assessing responses to genetic testing (e.g., decisional regret) that do not apply to untested individuals. Family history, eligibility of relatives, and willingness to invite relatives to the cohort are assessed with items used with cancer patients and at-risk individuals [18]. For this paper, respondents (index cases and relatives) are grouped based on whether they were willing to invite at least one relative to the cohort versus none. Predictors of intention to invite relatives were chosen according to the Theory of Planned Behavior (TPB) [19]. Demographics included age, gender, level of education, and marital status. Clinical characteristics included syndrome (HBOC versus LS), pathogenic variant identified (yes, no, untested), time since cancer diagnosis (never, ≤5 years, >5 years), and time since genetic testing (never, ≤5 years, >5 years).

Psychological characteristics included perceived cancer risk and genetic affinity. Perceived cancer risk was assessed with one item asking respondents to rate their chances of developing (another) cancer on a 11-point scale paired with verbal anchors, 1“Definitely will not” to 10 “Definitely will”. Genetic affinity, i.e., perceptions of being informed about cancer genetics was assessed by summing responses in two items asking: “How well informed do you feel about the probability of getting hereditary cancer?” ranging from 1 “Not at all informed” to 7 “Very Informed” and “How much do you know about the genetics of cancer?” ranging from 1 “Not at all” to 7 “A great deal”. The final score is created by averaging individual scores, with higher scores indicating higher genetic affinity.

Access to healthcare services was assessed with three items asking respondents to indicate whether they have a routine source of care (fewer than two providers organize their care); the type of healthcare provider they have seen most often the past 12 months (specialist versus general practitioner or family doctor); and whether there was a time during the past 12 months that they did not receive care due to high out-of-pocket costs.

An investigator-developed fifteen-item scale assessing family support at times of illness [18,20] was added to the model. Items ask respondents what applies to their family e.g., “In our family, when I have a health problem I can find someone to help me get to the doctor”. Items are scored on a 7-point Likert-type scale, ranging from 1 “Never True” to 7 “Always”. The final score is created by averaging individual scores, with higher scores indicating greater family support at times of illness. Cronbach’s alpha was 0.91 in this study.

## 3. Data Analyses

R software [21] was used for all statistical analyses. Significance was set at two-sided alpha = 0.05. The proportion of relatives invited to the cohort was calculated based on the total number of relatives each respondent intended to invite (numerator) divided by the total number of relatives potentially eligible for carrier testing (denominator). Two proportion z-test, Fisher’s exact test and Chi-square test assessed differences in frequencies and proportions.

Logistic regression was used to model the association of predictors with intention to invite relatives (at least one relative versus none). Missing values were less than 5% of data and no pattern was detected. Multiple imputations addressed missing values using the R software “mi” package [22]. For each copy of the imputed dataset (*n* = 5), variable selection was based on LASSO, using the R software “glmnet” package [23]. LASSO avoids issues related to multiple testing, reduces predetermined assumptions in the variable selection phase, and addresses multicollinearity problems when fitting statistical models [24]. Ten-fold cross validations determined the value of the tuning parameter λ in LASSO. The final model included predictors with non-zero coefficients in the fitted LASSO model. The means and standard errors of estimates for each imputed dataset were then pooled to create a combined estimate based on Rubin’s rules [25].

## 4. Results

Clinical sites identified 843 (607 HBOC and 236 LS) potentially eligible index cases (Figure 1). Among them, 730 (536 HBOC and 194 LS) were eligible for the CASCADE cohort and received an invitation letter. A signed consent was returned by 365 index cases (response rate 55.6%; 56.9% for HBOC versus 50.9% for LS, *p* = 0.20), after excluding index cases currently under recruitment (*n* = 70). From 304 index cases who had provided data at the time of this analysis, we identified 2,940 relatives, an average of 10 relatives per index case potentially eligible for cascade testing.

The proportion of index cases willing to invite at least one relative to the cohort was 64.8% (65.5% HBOC versus 62.1% LS, *p* = 0.71). We prepared 359 invitation letters asking 197 index cases to pass them on to relatives. Moreover, 60 relatives reported intention to invite 43 additional relatives (349 HBOC and 53 LS). Overall, this is an average of 1.6 invitations per respondent willing to invite at least one relative to the cohort. Response rate among relatives was 46.9% (*n* = 158), after excluding cases under recruitment (*n* = 45) and those not eligible (*n* = 20, e.g., not living in Switzerland). There was no difference in the proportion of consented relatives from the total invited between the two syndromes (47.1% HBOC versus 45.0% LS, *p* = 0.93). Common reasons for refusals among non-respondents were “I do not have enough time” and “I am not interested”.

Table 1 presents demographic and clinical characteristics of index cases and relatives by syndrome. HBOC respondents were significantly more likely to be female compared to LS respondents (90% HBOC versus 62% LS). A higher proportion of LS index cases were affected by cancer compared to HBOC index cases. Index cases (HBOC and LS combined) were also more likely than relatives to have a cancer diagnosis (20% vs. 64%, *p* < 0.01). HBOC cases were more likely to report seeing a specialist compared to LS index cases. Cascade testing was reported by 71 (61.7%) relatives, and this proportion was not different between the two syndromes. Genetic testing identified 54 relatives (47.0%) as carriers of the familial pathogenic variant, while 17 relatives (14.8%) were true negatives. The remaining 44 relatives (39.6% HBOC and 28.6% LS) were untested at the time they completed the baseline survey.

Approximately 45.1% of those who had genetic testing were tested less than 5 years ago. Among HBOC respondents, 144 reported carrying *BRCA1* variants and 117 reported *BRCA2* variants. Among LS respondents, 17 reported *MLH1* variants, 22 *MSH2* variants, 10 *MSH6* variants, and 6 *PMS2* variants. Based on answers from respondents, we estimated that approximately 18.2% of the non-invited relatives already had genetic testing. Considering relatives who had genetic testing but were not invited to the cohort, there were 1.5 carrier testing per index case. The first follow-up survey was completed by 211 (50.4%) individuals by December 2021. Data collection is ongoing, since not all respondents entered the cohort at the same time.

Table 2 presents the number of relatives potentially eligible for cascade testing by syndrome and by degree of relationship. Overall respondents were willing to invite less than 50% of their relatives to the cohort. Within each syndrome, there was higher intention to invite first-degree relatives (37.0% for HBOC and 35.4% for LS), compared to second- (18.6% for HBOC and 13.4% for LS) and third-degree relatives (14.3% for HBOC and 13.5% for LS). This finding was not different between the two syndromes. There was no difference in intention to invite relatives based on their gender, but HBOC respondents were more likely to invite a higher proportion of nieces compared to LS respondents.

We asked respondents identified as carriers of a pathogenic variant to indicate with whom they shared their genetic testing results. Responses were almost 100% consistent between the two syndromes. Approximately two out of three shared results with their spouse (61.3% HBOC and 54.8% LS), approximately half with their children (50.0% HBOC vs. 46.6% LS), while about one in three responded that they shared results with some of their relatives but not with all (37.7% HBOC vs. 37.0% LS). A smaller number responded that they would prefer not to communicate genetic test results with anyone in their family (6.6% HBOC vs. 13.1% LS). Approximately two out of three who did not share results provided reasons related to children (68.9% HBOC vs. 63.0% LS) i.e., “my children are too young”, “I don’t have any children”, “I don’t want to worry my children”. Fewer mentioned communication difficulties (13.2% HBOC vs. 37.0% LS, *p* = 0.03) i.e., “I find this to be very difficult”, “I have no contact with my relatives”, “I don’t know how to explain what the results of the test mean”, “I don’t know how to start the conversation” and “I didn’t have the opportunity”, or negative emotions as communication barriers (8.5% HBOC vs. 14.8% LS) i.e., “I feel guilty” and “I would rather not talk about things that make me upset or sad”.

We also asked carriers of pathogenic variants to indicate their degree of agreement with statements implying patient-mediated or provider-mediated communication of testing results to relatives. Figure 2 presents the proportion of individuals who responded “Somewhat Agree”, “Agree” or “Strongly Agree” to each statement. Approximately 50% would like to use a reliable website or a brochure to facilitate the communication process. Statements indicating provider-mediated communication were endorsed by approximately one in three or fewer carriers. Finally, joining a family support group was embraced by fewer than 20% carriers. Findings did not differ by syndrome.

Two separate logistic regression models examined predictors of intention to invite at least one relative to the cohort (209 HBOC and 48 LS) versus no intention to invite relatives (130 HBOC and 32 LS). Predictors are presented in the Appendix A along with results of the univariate analyses. Among the 15 potential predictors, univariate analyses showed that carrying a pathogenic variant was a significant predictor of intention to invite relatives for both syndromes. Additional predictors for HBOC were being diagnosed with cancer less than five years ago, being an index case, being older, and reporting higher genetic affinity and higher family support in times of illness.

Table 3 presents the final logistic regression models with six and two variables selected by LASSO as significant predictors of intention to invite relatives for HBOC and LS, respectively. The odds ratios indicate the multiplicative difference between intention and no intention to invite relatives. Carrying a pathogenic variant remained the most important predictor of intention to invite at least one relative to the cohort for both syndromes. The second predictor retained by LASSO in the fitted models was related to family environment, and was family support in illness for HBOC and being married or living as married for LS. LASSO also retained four additional predictors in the HBOC fitted model, i.e., being older, 12 years of education or higher, ever been diagnosed with cancer, and higher genetic affinity.

## 5. Discussion

The CASCADE cohort focuses exclusively on families harboring pathogenic variants associated with HBOC and LS, as opposed to families with multiple cancer cases but without a defined genetic status [26]. Recruitment takes place in diverse settings, which significantly increases sample representativeness. Switzerland includes several distinct cultural, linguistic, and geographical regions and populations within the same country and healthcare system, which adds significant value to our cohort [27]. The inclusion of both HBOC- and LS-harboring families in larger numbers compared to previous studies [28,29] enables between-syndrome comparisons.

Recruitment and retention rates are promoted through the study website (https://swisscascade.ch, accessed: 22 February 2022), annual newsletters, and an annual lottery. We calculated response rates of index cases and relatives in a conservative manner, excluding a considerable number who were in the recruitment window. Many relatives might have been invited by more than one respondent, which further decreases the response rate to the cohort. A response rate of 25–50% has been reported in other family-based cohorts [13,30]. Rates of carrier testing in relatives were similar to another European-based study [31], confirming that at least one in three at-risk individuals do not receive cascade testing. In contrast to others [28], we found no difference between the two syndromes in rates of relatives’ cascade testing, and our findings are based on a larger sample. We identified an average of 10 relatives per index case potentially eligible for cascade testing, which is within the range of 5–15 relatives reported by others and demonstrates the potential of cascade screening to impact public health [32,33]. An average rate of 1.5 genetic tests per index case is comparable to a Canadian study [34].

HBOC index cases were five times more likely to be female, consistent with evidence that males are less likely to be informed about the implications of HBOC for their own health [28]. Moreover, two out of three LS respondents were female, consistent with evidence that males with colorectal cancer are less likely to have genetic testing [35]. Although there was a slight preference towards inviting nieces among HBOC respondents, we did not find a significant gender difference in intention to invite relatives within each syndrome. Taken together, these findings suggest that although relatives of both genders are equally likely to be informed about hereditary cancer risk, males are less likely to act on this information and pursue cascade testing. This highlights the need for targeted interventions to address social stigma and increase awareness and cascade screening among males, especially among those potentially affected by HBOC [36].

Both HBOC and LS respondents were more willing to invite first-degree relatives to the cohort, with decreasing intention to contact relatives as the degree of relatedness increased. Carrying a pathogenic variant and the family environment were the most significant predictors of intention to invite relatives. Family disclosure is important because relatives can be alerted to their risk of carrying the pathogenic variant, increasing the feasibility, outreach, and yield of cascade screening programs [13,37,38]. In our sample, the majority of carriers wished to retain the responsibility for contacting relatives, possibly with the assistance of digital technology, and this finding was not different for the two syndromes. Statements indicating provider-mediated communication were embraced by fewer carriers, if they still maintained an active role in the communication process. Our findings are consistent with findings from an independent sample of Swiss women with LS [39], suggesting that in Switzerland, and possibly in other countries with similar attitudes towards privacy, efforts to improve cascade screening should be directed towards facilitating intra-familial communication. Efforts to expedite testing of relatives, e.g., digital educational programs, web-based counseling, and mailing of saliva kits [36,37], could be further explored for Swiss families. Digital interventions, which have been embraced by respondents, may help clarify the “black box” of cascade screening, e.g., easier access to information for relatives and tracking the number of invites sent by index cases and the number of relatives accepting the invitation. Expanding health insurance coverage for cascade testing to second- and third-degree relatives may facilitate cascade testing by removing structural barriers from the Swiss healthcare system [38]. Finally, our findings suggest that recommendations for the implementation of provider-mediated communication [39] may not be readily applicable in the Swiss context.

A limitation of the study is the smaller number of LS respondents, meaning that comparisons between the two syndromes should be interpreted with caution. Another potential limitation of the CASCADE cohort is that respondents may have better family relationships compared to non-respondents, which could present a selection bias. However, lack of intention to invite relatives to the cohort was not an exclusion criterion. Finally, refusal and non-responders do not threaten the validity of findings regarding carrier testing of relatives because there were multiple informants per family and multiple contacts per relative, similar to other family-based cohorts [40].

In conclusion, cascade screening for HBOC and LS is an important public health intervention for reducing cancer-related morbidity and mortality. Early identification of individuals with pathogenic variants allows adequate time for personalized prevention strategies and other interventions that enhance quality of life, also enhancing the effectiveness of cascade screening programs. Study findings can help optimize delivery of cancer genetic services in Switzerland and possibly in other countries.

## Figures and Tables

**Figure 1 cancers-14-01636-f001:**
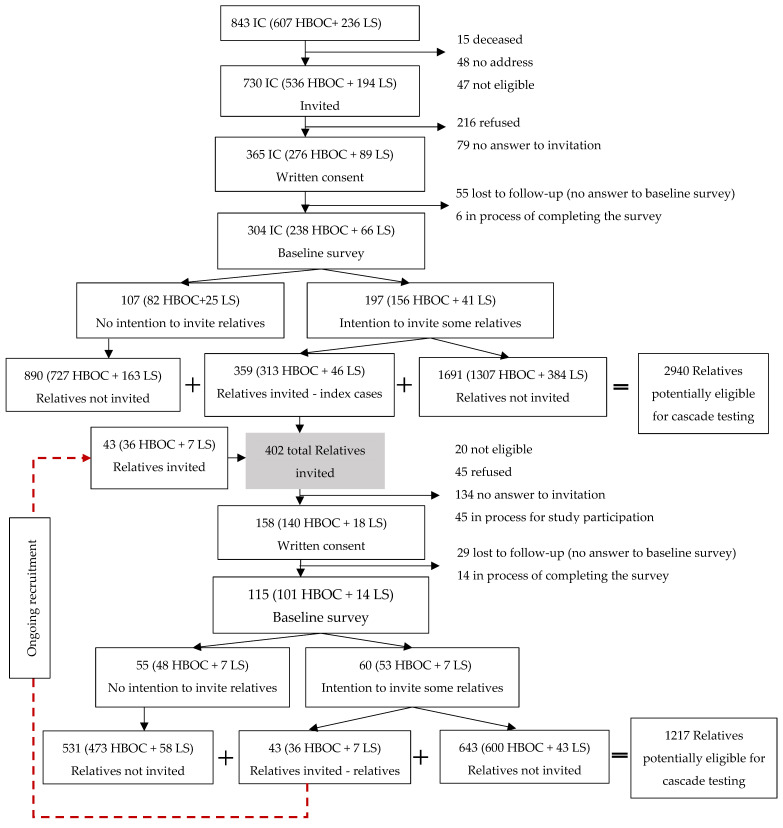
Consort diagram of recruitment process in the CASCADE cohort.

**Figure 2 cancers-14-01636-f002:**
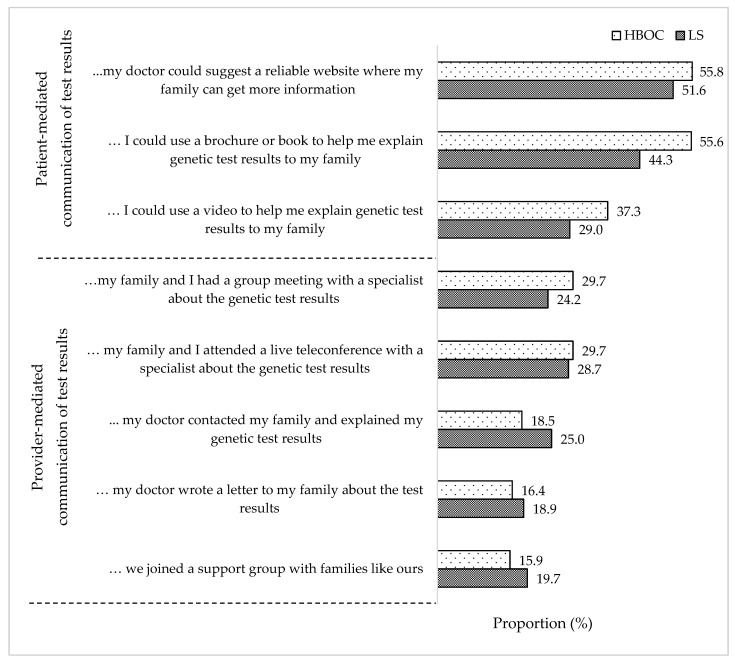
Preferences for provider involvement in family communication of test results for all carriers of familial pathogenic variants (index cases + relatives *n* = 358).

**Table 1 cancers-14-01636-t001:** Demographic and clinical characteristics of index cases and relatives by syndrome.

Index Cases	Total *n* = 304 (% or ± SD)	HBOC *n* = 238 (% or ± SD)	LS *n* = 66 (% or ± SD)	HBOC vs. LS *p*-Values ^a^
Gender	F = 255 (83.9)	F = 214 (89.9)	F = 41 (62.1)	**<0.01**
Caucasian origin	254 (83.6)	199 (83.6)	55 (83.3)	1
Age	52.3 (±12.6)	52.1 (±12.2)	52.9 (±14.0)	0.67
Education level (12th years of education or higher)	258 (84.9)	206 (86.6)	52 (78.8)	0.17
Married or living as married	224 (73.7)	183 (76.9)	41 (62.1)	**0.02**
Cancer affected ^c^	195 (64.1)	145 (60.9)	50 (75.8)	**0.04**
Breast cancer (invasive)		95 (39.9)	3 (4.5)	-
Ovarian cancer		29 (12.2)	4 (6.1)	-
Prostate cancer		1 (0.4)	2 (3.0)	-
Pancreatic cancer		0	0	-
Colon or rectal cancer		2 (0.9)	30 (45.5)	-
Endometrial cancer		5 (2.1)	8 (12.1)	-
Other ^d^		49 (20.6)	13 (19.7)	-
Age at 1st cancer diagnosis	47.7 (±11.7)	47.4 (±11.7)	48.3 (±11.7)	0.80
Genetic testing ≤5 years ago	137 (45.1)	108 (45.4)	29 (43.9)	0.95
Has a routine source of care (ref: >2 healthcare professionals)	288 (94.7)	223 (93.7)	65 (95.5)	0.22
Provider seen most often is specialist (ref: generalist)	175 (57.6)	150 (63.0)	25 (37.9)	**<0.01**
Out of pocket cost is a barrier to accessing care	18 (5.9)	17 (7.1)	1 (1.5)	0.14 ^b^
**Relatives**	**Total n = 115** **(% or ± SD)**	**HBOC n = 101** **(% or ± SD)**	**LS n = 14** **(% or ± SD)**	**HBOC vs. LS** ***p*-values ^a^**
Gender	F = 78 (67.8)	F = 69 (68.3)	F = 9 (64.3)	1
Caucasian origin	108 (93.9)	94 (93.1)	14 (100)	0.67
Age	49.7 (±16.9)	50.4 (±17.2)	44.6 (±14.3)	0.19
Education level (12th years of education or higher)	104 (90.4)	91 (90.1)	13 (92.9)	1
Married or living as married	84 (73.0)	78 (77.2)	6 (42.9)	0.34 ^b^
Cancer affected	23 (20.0)	19 (18.8)	4 (28.6)	0.50 ^b^
Breast cancer (invasive)		13 (12.9)	0	-
Ovarian cancer		3 (3.0)	2 (14.3)	-
Prostate cancer		1 (1.0)	0	-
Pancreatic cancer		0	0	-
Colon or rectal cancer		0	1 (7.1)	-
Endometrial cancer		0	0	-
Other ^d^		7 (6.9)	2 (14.3)	-
Age at 1st cancer diagnosis	48.4 (±11.1)	52.7 (±12.8)	46.4 (±8.8)	0.47
Had genetic testing	71 (61.7)	61 (60.4)	10 (71.4)	0.82 ^b^
Familial pathogenic variant identified	54 (47.0)	47 (46.5)	7 (50.0)	1 ^b^
Familial pathogenic variant excluded	17 (14.8)	14 (13.9)	3 (21.4)	0.46 ^b^
Genetic testing ≤5 years ago	32 (27.8)	28 (27.7)	4 (28.6)	1 ^b^
Has a routine source of care (ref: >2 healthcare professionals)	104 (90.4)	93 (92.1)	11 (78.6)	0.26
Provider seen most often is specialist (ref: generalist)	45 (39.1)	43 (42.6)	2 (14.3)	0.24 ^b^
Out of pocket cost is a barrier to access care	5 (4.3)	3 (3.0)	2 (14.3)	0.13 ^b^

^a^ Two proportion z-test; ^b^ Fisher’s exact test; ^c^ The number of cancer diagnoses is larger than the number of individuals affected with cancer because some respondents reported more than one diagnosis; ^d^ Includes brain, lung, stomach, gallbladder, liver, small intestine, kidney, urinary tract, and cervix; Abbreviations: F, female; SD, standard deviation; Bold = significant two-tailed *p* value ≤ 0.05.

**Table 2 cancers-14-01636-t002:** Relatives potentially eligible for genetic testing versus proportion of relatives that respondents intend to invite by degree of relationship and syndrome.

Degree of Relationship		HBOC		LS		HBOC vs. LS ^a^
		Eligible (n)	Intend to Invite (n, %)	Eligible (n)	Intend to Invite (n, %)	p (Intend to Invite)
**First-degree**	Daughters	173	76 (43.9%)	42	18 (42.9%)	1
	Sons	175	86 (49.1%)	38	15 (39.5%)	0.37
	Sisters	235	78 (33.2%)	40	14 (35.0%)	0.97
	Brothers	215	84 (39.1%)	49	18 (36.7%)	0.89
	Parents	306	85 (27.8%)	71	20 (28.2%)	1
	Sum	1104	409 (37.0%)	240	85 (35.4%)	0.69
Second-degree	Granddaughters	39	13 (33.3%)	5	0	0.07 ^c^
	Grandsons	33	8 (24.2%)	10	2 (20.0%)	1 ^b^
	Nieces	**174**	**43 (24.7%)**	**39**	**2 (5.1%)**	**0.02 ^b^**
	Nephews	214	47 (22.0%)	39	7 (17.9%)	0.83 ^b^
	Half-sisters	23	3 (13.0)	7	2 (28.6%)	0.59 ^b^
	Half-brothers	16	3 (18.8%)	10	2 (20.0%)	1 ^b^
	Aunts	322	47 (14.6%)	60	6 (10.0%)	0.54 ^b^
	Uncles	283	41 (14.5%)	61	10 (16.4%)	0.70 ^b^
	Sum	1104	205 (18.6%)	231	31 (13.4%)	0.08
Third-degree	Female cousins	607	98 (16.1%)	101	13 (12.9%)	0.49
	Male cousins	641	80 (12.5)	129	18 (14.0%)	0.75
	Sum	1248	178 (14.3%)	230	31 (13.4%)	0.83
Total		3456	792 (22.9%)	701	147 (21%)	0.88
Between Degree of Relationship *P (Sum)* ^d^		**<0.01**		**<0.01**		
Between Gender *P (Sum)* ^d^		0.36		0.90		

^a^ Two proportion z-test; ^b^ Fisher’s exact test; ^c^ binomial test; ^d^ Chi-square test; Bold = significant two-tailed *p* value ≤0.05.

**Table 3 cancers-14-01636-t003:** Logistic regression fitted models of intention to invite at least one relative versus intention to invite no relatives for HBOC and for LS based on least absolute shrinkage and selection operator (LASSO) variable selection.

Predictors		HBOC		LS	
		OR	95% CI	OR	95% CI
**Demographic Characteristics**	Age	1.01	1.00–1.03	-	-
	<12 years of education (reference group)	-	-		
	High school graduate	0.84	0.35–1.98	-	-
	University degree or higher	0.61	0.25–1.48	-	-
Clinical Characteristics	Carrying a pathogenic variant (ref: no variant or untested)	**3.03**	**1.68–5.46**	**4.53**	**1.20–17.10**
	No cancer	-	-		
	Cancer diagnosis <5 years	1.50	0.70–3.23	-	-
	Cancer diagnosis >5 years	0.81	0.45–1.44	-	-
Knowledge and Attitudes	Genetic affinity	1.14	0.93–1.39	-	-
Family environment	Married or living as married (ref: single or living alone)	-	-	2.03	0.81–5.09
	Family Support in Illness	**1.32**	**1.01–1.73**	-	-

Bold: Statistically significant.

## Data Availability

The CASCADE Consortium is open to collaborations with national and international researchers. We invite interested parties to contact the study team through website (https://swisscascade.ch/en/contact-2/, accessed on 22 February 2022) to discuss project ideas, data access, and the submission of research concepts to the Scientific Board. Templates for data requests and further information on the study are available (https://swisscascade.ch/en/research-project-data-request/, accessed on 22 February 2022).

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
