# Peer review of "Intention to Inform Relatives, Rates of Cascade Testing, and Preference for Patient-Mediated Communication in Families Concerned with Hereditary Breast and Ovarian Cancer and Lynch Syndrome: The Swiss CASCADE Cohort"

_cancers, 2022, doi:10.3390/cancers14071636_

Round 1

Reviewer 1 Report

The clinical outcome and treatment of patients should be discussed in this manuscript. Will other homologous recombination or mismatch repair genes be analyzed. Also more information about the low response rate of children and grandchildren should be discussed. What will the the treatment outcome of these patients.

Reviewer 2 Report

Thank you to the authors for this interesting investigation into uptake of cascade genetic testing in families with hereditary cancer syndromes. This study provides some insight into areas of clinical improvement.

Approximately 40% of the relatives identified and consented to the study have not had genetic testing. Is there scope to provide some indication as to why they have chosen not to proceed with testing?

The authors have used the word "embrassed" on three occasions (lines 271, 339, 347) when I believe the word "embraced" should be substituted? 
